# Combined Hyperthermia and Re-Irradiation in Non-Breast Cancer Patients: A Systematic Review

**DOI:** 10.3390/cancers15030742

**Published:** 2023-01-25

**Authors:** Ji-Young Kim, Sebastian Zschaeck, Jürgen Debus, Fabian Weykamp

**Affiliations:** 1Department of Radiation Oncology, Heidelberg University Hospital, 69120 Heidelberg, Germany; 2Heidelberg Institute of Radiation Oncology (HIRO), German Cancer Research Center (DKFZ), 69120 Heidelberg, Germany; 3Department of Radiation Oncology, Charité - Universitätsmedizin Berlin, 10178 Berlin, Germany; 4Berlin Institute of Health (BIH), Charité - Universitätsmedizin Berlin, 10178 Berlin, Germany; 5Department of Radiation Oncology, Heidelberg Ion-Beam Therapy Center (HIT), Heidelberg University Hospital, 69120 Heidelberg, Germany; 6Clinical Cooperation Unit Radiation Oncology, German Cancer Research Center (DKFZ), 69120 Heidelberg, Germany; 7German Cancer Consortium (DKTK), Partner Site Heidelberg, 69120 Heidelberg, Germany

**Keywords:** hyperthermia, re-irradiation

## Abstract

**Simple Summary:**

Hyperthermia in the range of 39–43 °C is associated with potent radiosensitization. The clinical benefit of combined radiotherapy and hyperthermia has been shown for a variety of indications. Regarding re-irradiation, added hyperthermia can be considered for locoregional breast cancer recurrence with isolated unresectable disease or as an adjuvant therapy after resection. For other cancer entities, the clinical evidence is scarce. This review aims to systematically summarize the currently available literature. The final search was performed on 29 August 2022.

**Abstract:**

Purpose: This systematic literature review summarizes clinical studies and trials involving combined non-ablative hyperthermia and re-irradiation in locoregionally recurrent cancer except breast cancer. Methods: One database and one registry, MEDLINE and clinicaltrials.gov, respectively, were searched for studies on combined non-ablative hyperthermia and re-irradiation in non-breast cancer patients. Extracted study characteristics included treatment modalities and re-irradiation dose concepts. Outcomes of interest were tumor response, survival measures, toxicity data and palliation. Within-study bias assessment included the identification of conflict of interest (COI). The final search was performed on 29 August 2022. Results: Twenty-three articles were included in the final analysis, reporting on 603 patients with eight major tumor types. Twelve articles (52%) were retrospective studies. Only one randomized trial was identified. No COI statement was declared in 11 studies. Four of the remaining twelve studies exhibited significant COI. Low study and patient numbers, high heterogeneity in treatment modalities and endpoints, as well as significant within- and across-study bias impeded the synthesis of results. Conclusion: Outside of locoregionally recurrent breast cancer, the role of combined moderate hyperthermia and re-irradiation can so far not be established. This review underscores the necessity for more clinical trials to generate higher levels of clinical evidence for combined re-irradiation and hyperthermia.

## 1. Introduction

Hyperthermia in combination with radiotherapy has demonstrated clinical benefit in terms of tumor response, survival and palliation with acceptable added toxicity in a variety of indications [1,2,3,4,5]. Hyperthermia in the range of 39–43 °C leads to potent radiosensitization [6,7,8]. Moreover, the cytotoxic effects as well as the thermal enhancement of radiotherapy conferred by hyperthermia have been shown to be particularly effective in conditions associated with radioresistance [9,10].

No standard of care exists for locoregional recurrence in most tumor types, in particular for those patients with intense prior treatments where concerns about acquired resistance and dose-limiting toxicity are high [11]. For patients who have undergone previous radiotherapy, adjuvant hyperthermia in addition to re-irradiation may allow for enhanced local tumor control.

Effective hyperthermia treatment requires specialized heating equipment, treatment planning, precise thermometry and adequate quality assurance. The types of hyperthermia delivery systems under clinical use have been reviewed elsewhere [12,13]. Briefly, the delivery systems can be divided in terms of heated volume (local, regional, whole-body), depth (superficial vs. deep) and applicator position (internal vs. external).

The tumor location is decisive in selecting the type of hyperthermia applicator, which has resulted in the development of dedicated heating equipment for specific tumor sites and anatomical locations. Importantly, these devices vary considerably in terms of physical energy transduction, delivery and control.

Most of the studies included in this review employ electromagnetic heating, which can be further subdivided into categories depending on the frequency of the generated electromagnetic field. In ascending order of frequency and decreasing in tissue penetration depth, these categories include radiofrequency (RF, ~3 Hz–300 MHz), microwave (MW, ~300 MHz–300 GHz) and infrared (IR, ~300 GHz–430 THz) [13]. In internal hyperthermia, energy is applied to a limited tissue volume close to the interstitial, intraluminal or intracavitary applicator. As such, interstitial HT is usually combined with brachytherapy. For more deeply seated (>2 cm from the skin surface) tumors, external heating at depth can be achieved by capacitive systems or radiative systems; in the latter case, heating is usually accomplished by employing a circumferential phased-array approach in which multiple antennas are arranged around the patient to generate a constructive wave interference to reach the temperature goal in the target region [12].

The pleiotropic effects on normal and cancerous tissue attributed to locoregional hyperthermia have previously been reviewed elsewhere [14]. Mild hyperthermia from 39 °C to 42 °C leads to increased blood flow and oxygenation [15], although increased exposure to even higher temperatures can restrict blood flow [16]. However, direct cell-killing effects mediated by hyperthermia increase exponentially from 41 °C onwards [17]. Inhibition of homology-directed DNA-damage repair, associated with heat-induced degradation of BRCA2 [18], strongly increases at temperatures above 41 °C. Fever-range hyperthermia is a potent regulator of innate and adaptive immunity [19]. The addition of hyperthermia to radiotherapy has been associated with higher necrosis rates [20,21], release of DAMPs [22] such as HSP70 and HMGB1, infiltration of dendritic cells [23], and NK-cell activation [24] and cytotoxicity [25].

Even though clinical hyperthermia aims for a target temperature of 41–43 °C, anatomical and technical variables lead to higher variations of attained temperatures between 39 and 45 °C [26]. Nonetheless, it has been argued that heterogeneous temperatures during hyperthermia sessions may yield complementary immunological effects, such as increased blood flow and immune cell infiltration at 39 °C and enhanced immunogenic cell death rates at temperatures higher than 41 °C.

The hyperthermia-induced radiosensitization observed in both pre-clinical and clinical studies can therefore be attributed to synergistic effects, such as the oxygenization of radioresistant, hypoxic niches; inhibition of DNA repair; and immunomodulation.

Corroborating clinical evidence of combined hyperthermia and re-irradiation has been shown in breast cancer [2,4] and has been summarized in previous reviews and a meta-analysis [27,28]. The greatest benefit in tumor control was observed in patients undergoing re-irradiation, for whom the irradiation doses had to be limited. Other systematic reviews have identified prognostic factors for outcome which may aid in patient selection [29,30]. Interestingly, a more recent retrospective study demonstrated effective tumor control with further reduction in irradiation dose combined with superficial hyperthermia [31], allowing for multiple re-treatments.

According to the German expert guideline for gynecological oncology (AGO), hyperthermia and re-irradiation can be recommended in locoregional breast cancer recurrence with isolated unresectable disease or as adjuvant re-irradiation after resection [32].

For the remaining cancer types besides breast cancer, far fewer clinical studies have been conducted for combined non-ablative hyperthermia and re-irradiation. This review therefore aims to summarize the currently available clinical evidence and thereby aid in providing suitable indications for further studies.

## 2. Materials and Methods

The systematic review was conducted following the 2020 updated PRISMA guidelines [33]. Studies were searched using the MEDLINE database, via the freely accessible PubMed interface, and the registry clinicaltrials.gov from inception to 29 August 2022 (Figure 1).

In PubMed, the following search query was used: (‘Hyperthermia, Induced’[Mesh] OR hyperthermia[tw] OR heat[tw] OR thermotherapy[tw] OR thermoradiotherapy[tw]) AND (reirradiat*[tw] or re-irradiat*[tw] or (recurrent[tw] AND radiotherapy[tw])) NOT (Review [Publication Type] OR "breast cancer"[Title] OR "HIFU"[tw] OR "HIPEC"[tw]). 

On clinicaltrials.gov (last accession date: 29 August 2022), studies were searched with the condition “recurrent cancer” and the intervention “hyperthermia, radiotherapy”. The search was not limited to any date. Only English articles were considered.

Single-arm, double-arm, retrospective and prospective studies (randomized and non-randomized) fulfilling the following criteria were included: patients with in-field cancer recurrences, persistent cancer or second primary cancers undergoing combined re-irradiation and hyperthermia.

After the exclusion of duplicates, articles were screened according to their titles and abstracts. The following article types were excluded:-Duplicates;-Articles that were not clinical studies;-Studies on breast cancer;-Articles that were updated in a later publication by the same author(s);-Studies involving fewer than 10 patients with one tumor type treated with re-irradiation and hyperthermia;-Case reports, conference abstracts or presentations.

Endpoints of interest included those related to tumor response and clinical outcome, such as complete/partial response (CR/PR), duration of local control (DLC), overall survival (OS) and progression-free survival (PFS), as well as reported treatment-related toxicities, in particular thermal toxicity (i.e., thermal blisters). Patient characteristics and information regarding prior and recurrence treatments were extracted. Re-irradiation dose concepts and hyperthermia delivery systems were listed. Studies were stratified according to major tumor groups. Treatment concepts as well as outcome and toxicity data were compared between the studies within the group and, if appropriate, to comparable studies in the literature.

Bias factors in individual studies were assessed by evaluating potential conflicts of interest (i.e., patent applications) and the provided funding statements.

The literature search strategy and bias assessment were performed independently by two authors (J.Y.K. and F.W.) and cross-validated. Disagreements were resolved in discussion.

## 3. Results

### 3.1. Study Selection, Patient and Treatment Characteristics 

From an initial yield of 276 articles and 32 registered clinical trials, 23 studies were included in the final analysis (Table 1). Seventy-one full-text records were assessed for eligibility. The most frequent reason for exclusion was unclear patient characteristics, as multiple studies reported aggregated data which precluded extraction of the group of interest (number of patients with defined tumor type undergoing re-irradiation and hyperthermia).

These 23 studies report on 603 patients who underwent treatment with combined hyperthermia (HT) and re-irradiation (re-RT) for eight major tumor types. The median number of relevant patients from the studies included in this review was 16.

The hyperthermia treatment modality naturally depended on the tumor entity. For example, studies employed interstitial HT for HN and gynecological malignancies.

Twelve out of twenty-three studies were retrospective analyses. Ten of the remaining eleven prospective studies were single-arm trials or included within a larger cohort a fraction of patients undergoing re-RT and/or additional HT treatment. Only one trial from 1996 was a randomized and controlled phase III study.

Due to the small numbers of patients in the studies included in this review, pre-therapeutic prognostically relevant parameters are likely to represent significant confounding variables. Patient characteristics, interventions and endpoints showed a high degree of heterogeneity, which precluded a meta-analysis of observed therapeutic effects.

### 3.2. Bias Assessment

Eleven of twenty-three studies (47.8%) did not declare COI statements. Funding statements were missing in 14 of 23 studies (61%). Four of the twelve studies (33.3%) with COI statements exhibited relevant COI in relation to the presented work. In all four studies, the first or senior authors founded companies focused on developing the respective hyperthermia technology for clinical use. Three of the four studies declared at least partial funding from those companies.

### 3.3. Head and Neck Cancer

Nine studies [34,35,36,37,38,39,40,41,42,43] have been included in this analysis (Table 2). Five studies report on combined interstitial radiotherapy and hyperthermia, in which case interstitial hyperthermia was performed (ITRT).

Phase I/II studies in the 1980s and 1990s [57,58] had shown superior CR and LC rates in patients with recurrent HN cancer treated with ITRT compared to historical controls. However, in a prospective, randomized phase III study by Emami et al. from 1992 comparing ITRT directly to IRT, added hyperthermia did not lead to a statistically significant effect in terms of tumor response, LC or survival [35]. However, upon setting minimum adequacy criteria for HT delivery, only one patient in the study was shown to meet the minimum criteria, indicating that the lack of QA guidelines and compliance for hyperthermia prohibits analysis of its clinical effects. Higher acute grade 4 toxicity concerning skin and mucosal reactions was observed in the ITRT group.

Two more studies included in this review, by Puthawala et al. [37] and Bartochowska et al. [40], did not find a significant effect of hyperthermia on clinical outcome in patients treated with interstitial brachytherapy, although Puthawala et al. mention, albeit without data, better locoregional control upon added hyperthermia treatment.

Two studies investigated triple combination therapy. The study by Feyerabend et al. [36] employed triple therapy with hyperfractionated RT, cisplatin chemotherapy and weekly MW hyperthermia for patients with high-risk lesions which resulted in a low CR rate of 8% and early termination of the study. Another study by Geiger et al. primarily showed that triple therapy PDR-BT, cisplatin and 5FU chemotherapy and interstitial hyperthermia was feasible with acceptable toxicity. However, in a retrospective analysis published by the same group, added hyperthermia was not shown to influence clinical outcome, unlike chemotherapy [59].

Superficial MW hyperthermia and re-irradiation were explored in one study by Gabriele et al. in 2009 for 14 patients with superficial HN recurrences. Treatment compliance was high with 93% of all planned hyperthermia sessions completed and no reported toxicities of grade 3 or higher. CR was shown in 33.3% of patients, and PR was shown in 25%.

Two more recent studies explore the feasibility of combined locoregional hyperthermia and re-irradiation using the Hypercollar [41] or its successor, the Hypercollar3D [42] device. These applicators were developed specifically for deep-seated recurrent or second primary HN cancers. Inverse treatment planning is performed by generating a specific absorption rate (SAR) pattern based on a 3D model derived from the RT-planning CT and subsequent optimization [60]. These studies are co-authored by the founders of Sensius BV, from which partial funding was received for both studies. Verduijn et al. [41] report 2-year LC and OS rates of 36% and 33%, which were corroborated by the later study by Kroesen et al., with rates of 37% and 53%, respectively. In comparison, patients with recurrent HN cancer treated with definitive re-irradiation with or without chemotherapy report 2-year LC and OS rates of 43% and 36%, respectively [61].

While the studies by Verduijn et al. [41] and Kroesen et al. [42] both claim high treatment compliance, a high fraction of acute grade 2 trismus in initial patients treated according to the protocol by Verduijn et al. necessitated a protocol revision in the study by Kroesen et al.

Fever-range whole-body hyperthermia (FRWBH) combined with re-irradiation in 10 patients with recurrent HN cancers was investigated by Zschaeck et al. [43] in a prospective phase I study. However, the study failed to meet the feasibility endpoint, with only 5 out of 10 patients receiving all required HT cycles, due to poor patient compliance, acute infections during treatment, claustrophobia and COVID-19 pandemic restrictions.

In conclusion, despite previous reports on the significant effect of added hyperthermia versus radiotherapy alone in patients with HN cancer [62], the potential of hyperthermia-induced radiosensitization has not been conclusively mirrored in clinical studies in patients with prior irradiation in the head and neck region.

In a conference poster excluded from this review, Yang et al. [63] report preliminary data on 33 patients with unresectable and recurrent head and neck cancer with previous irradiation in a single-arm phase II study treated with concurrent chemoradiotherapy and weekly hyperthermia. In this yet non-peer-reviewed contribution, Yang et al. report a CR rate of 57.6%, PR of 24.2%, 2-year OS rate of 61.9%, median PFS of 13.1 months, and 1-year LRR and DM rates of 39.3% and 13%, respectively. No grade 3 or higher skin or mucosal toxicities were observed. Grade 3–4 hematologic toxicity was reported in 9.1% of patients, and osteoradionecrosis was reported in 30.3% of patients.

### 3.4. Glioma

Three studies [47,48,49] investigating concurrent hyperthermia and re-irradiation in CNS malignancies were included (Table 3).

The studies by Maier-Hauff et al. recruited patients with recurrent glioblastoma (GBM) and employed iron oxide nanoparticles as the hyperthermic agent, using neuronavigationally controlled deposition [47,48]. Heat was then generated by subjecting the particles to an external alternate magnetic field (MFH-300F (NanoActivator, MagForce AG, Berlin, Germany). The authors include co-founders of MagForce AG and maintain patents on the use of their proprietary nanoparticle hyperthermic treatment in cancer therapy. Furthermore, the phase II study by Maier-Hauff et al. [48] reports funding by MagForce AG.

Regarding outcome, the phase II study by Meier-Hauff et al. reports a median OS of 13.4 months from the day of recurrence diagnosis. The EORTC-NCIC trial [64] shows a median survival after progression of 6.2 months for patients initially treated with combined RT and temozolomide (TMZ), among which 39% however only received best supportive care. The meta-analysis by Wong et al. [65] reports a median survival of 5.8 months for patients treated with chemotherapy upon recurrence. However, a retrospective analysis of 198 patients in the Charité by Kaul et al. [66] including 32 patients with recurrent GBM who were treated with nanoparticle therapy and re-irradiation during the time of the two-center (one of which was the Charité) phase II study by Maier-Hauff from 2007 to 2009 reports a median survival (defined as the time from the first day of re-irradiation onwards) of 6 months, similar to that reported for patients treated without nanoparticle therapy. Nanotherapy was not significantly associated with a better outcome.

The reported thermal dose parameters in the nanoparticle studies show mean Tmax values far above the degree span predictive of the biological effects conferred by hyperthermia. In contrast, estimated tumor temperature coverage remains low and CEMT43C90 values vary widely between the patients. These results indicate that the necessity remains for technical optimization for hyperthermia delivery.

Concerning toxicity, the larger phase II study by Maier-Hauff et al. [48] with 65 patients for whom toxicity data are available reports seizures and motor disturbances in one out of four to five cases.

The study by Heo et al. [49] employed capacitive hyperthermia with external thermometry probes and included a large fraction (80%) of patients with recurrent grade III glioma, rather than GBM. This study reports a median OS of 8.4 months, compared to the 7.9–11 months from studies investigating FSRT for patients with recurring high-grade glioma [67] or GBM only [68] and additional TMZ [69,70], all of which contain cohorts with a significantly higher median KPS score. However, the large fraction of patients with grade III glioma in the study by Heo et al. [49] may have significantly contributed to the outcome. No toxicities of grade 3 or higher were reported.

Analysis of the results above warrants further studies on combined hyperthermia and re-irradiation in patients with recurrent high-grade glioma. While the reported rate of initial tolerability is high, there are yet insufficient data and technical capabilities in both iron oxide nanoparticle and external (capacitive) hyperthermia combined with radiotherapy to demonstrate clinical benefit and safety in patients with recurrent glioma.

### 3.5. Radiation-Associated Sarcoma

Three studies [50,51,52] exploring combined hyperthermia and re-irradiation for radiation-associated sarcoma were identified (Table 4). All three presented studies deal with RAS in the thoracic region, and only the study by de Jong et al. [50] includes patients with histology other than angiosarcoma (5 out of 16 patients with non-(lymph-)angiosarcoma).

Radiation-associated angiosarcoma of the breast (RAASB) occurs in 0.1% of all BC patients after RT with an average latency of 8 years [71,72,73]. In comparison to other RASs and primary angiosarcoma [74], RAASB is characterized by recurrent c-myc amplification, a shorter latency from prior RT and possibly better treatment response [75]. Despite the lack of a uniform treatment recommendation, the standard of care aims for surgery with R0 resection, as radical resection determines LC and OS [73,76].

The studies by de Jong et al. [50] and Linthorst et al. [51] utilized MW hyperthermia and mean re-RT doses of 32–35 Gy. The study by Notter et al. employed water-filtered infrared-A (wIRA) hyperthermia, and most patients were treated with 20 Gy/4 Gy, with a similar concept to the authors’ recent publication on combined re-RT and HT in locoregional recurrent breast cancer [31].

The studies by de Jong et al. [50] and Lindhorst et al. [51] report median OSs of 9 and 12 months, respectively. For patients with unresectable RAS, the 3-year LC rates are 31% and 22%, respectively. In the case of adjuvant re+RT + HT, Lindhorst et al. report a 3-year LC rate of 46%. Concerning toxicity, de Jong et al. [50] and Lindhorst et al. [51] report grade 3 or higher toxicities in 6.25% (one case of grade 4 peripheral limb ischemia) and 8.7% of patients (one osteoradionecrosis case and one chronic wound case), respectively.

The study by Notter et al. [52] reports on ten patients with RAASB with four subgroups divided by the respective re-RT-indication. The median OS is 17 months. Interestingly, two patients were treated with the same therapy scheme for re-recurrences, each time resulting in CR. No grade 3 or higher toxicities were reported.

Combined re-RT and surgery are explored in single-institution retrospectives by Palta et al. [77] and Scott et al. [78] in which hyperfractionation and accelerated re-RT (HART) with total doses up to 75 Gy were applied. The 5-year OS rates and 5-year LC rates were reported to be 86% and 75%, as well as 64% and 92%, respectively.

The outcome data from the re-RT + HT studies included in this review fare well below the HART studies, even though significant differences in the patient collective are possible, given the scarcity of studies available.

Re-RT + HT can at this point be considered as (neo)adjuvant therapy or in case of unresectability. The possibility of re-treatments is promising, as presentation with re-recurrences in patients with RAASB is frequent. At this point, however, there is no study demonstrating clinical benefit conferred by added hyperthermia. Future studies may directly compare re-RT vs. re-RT + HT.

### 3.6. Rectal Cancer

Three studies [44,45,46] were identified for combined re-irradiation and hyperthermia in rectal cancer (Table 5).

Juffermans et al. [44] published a single-institution retrospective analysis on 54 patients treated with a median re-RT dose of 32Gy and four weekly HT sessions per patient. The primary endpoints concerned palliative effect, which was reported in 83% of patients with a median duration of six months. Treatment compliance was high, and no severe toxicities were observed.

Both Milani et al. [45] and Ott et al. [46] explored triple therapy consisting of chemoradiotherapy combined with regional hyperthermia. In a single-institution prospective phase I/II study by Milani et al. [45], 24 patients were treated with a median re-RT dose of 39.6 Gy in combination with 5FU, as well as biweekly HT sessions. A significant fraction of the patients had locally extensive disease with bone infiltration, and no patient underwent subsequent surgical resection. The primary endpoint, 3-year LPFS, was 15%. The actuarial 3-year OS was 30%. Palliation was observed in 50% of all patients and in 70% of patients with tumor response. Severe toxicity was reported in three patients (12.5%) with grade 3 diarrhea, which was persistent in one patient as late toxicity.

Ott et al. [46] report on data on the HyRec trial which included 16 patients with locally recurrent rectal cancer, 10 of which had a history of prior pelvic irradiation. As the aim was to increase the rate of curative resections, patients were treated with a relatively intense treatment protocol, consisting of simultaneous chemotherapy with 5FU and oxaliplatin, bi-weekly HT sessions and a re-irradiation dose of 45 Gy/1.8 Gy. Treatment adherence for both RT and HT was high. Sixty-three percent of LRRC patients underwent curative surgery. Pathological complete response (pCR) rates were 19% for both LARC and LRRC patients. The 3-year OS rate of LRRC patients was reported to be 85%.

In all three trials, treatment completion rates for both re-RT and HT were high. In light of recent developments in the field regarding organ preservation, intensification of local neoadjuvant treatment with the addition of hyperthermia may be a viable option to further suppress local recurrence [79,80]. However, given the low number of LRRC patients undergoing curative resection in the study by Ott et al. [46], further studies are required to confirm the high observed pCR rate.

### 3.7. Other Cancer Types

Two studies [53,54] were identified that investigated combined ITRT for in-field recurrences of gynecological malignancies in patients with a history of pelvic irradiation (Table 6).

The 21-patient (12 with recurrent cervical cancer) cohort of Surwit et al. [53] was treated with a median dose of 22 Gy, applied via LDR-BT, combined with a single 30 min session of iHT. CR and PR rates were 33% and 48%, respectively, with median durations of 11 and 3 months. Subjective pain relief was reported in 13 out of 17 responding patients.

Gupta et al. [54] report on 69 patients with pelvic gynecological recurrences. Among these, 15 patients (10 patients with recurrent endometrial cancer) had a history of pelvic irradiation and were treated with a modified Martinez universal perineal interstitial template (MUPIT) to accommodate both interstitial BT and iHT. They report a 3-year LC rate of 49%. Grade 4 toxicities were reported for 14% of all patients. HT was not significantly associated with LC duration or toxicity.

Even though the addition of hyperthermia has been associated with an improved response without added toxicity when added to chemoradiotherapy in the treatment of locally advanced cervical cancer [81,82], a definite role for hyperthermia in recurrence/re-irradiation treatment has not been formulated or demonstrated.

The study by Yamaguchi et al. [55] employed re-irradiation with 3D-CRT in 31 patients with persistent or recurrent esophageal cancer who had undergone prior radiotherapy as part of their primary treatment. Fourteen patients were treated with additional RF-capacitive hyperthermia. No significant effect on survival was observed with HT.

Ohguri et al. [56] report a single-institution retrospective study on 33 patients treated with combined re-irradiation and RF-capacitive hyperthermia for locoregionally recurrent NSCLC. The RT completion rate was 97% and the median OS was 18.1 months.

Two other studies in the literature have explored monotherapy with re-irradiation in NSCLC patients with locoregional recurrence and a similar median re-RT dose [83,84]. These reported a lower median OS of 7–8 months.

However, as this represents only a singular retrospective case series, more studies are required to substantiate a potential survival benefit conferred by added hyperthermia.

## 4. Discussion

Patients are referred for re-irradiation in unresectable cases or upon failure of other treatment modalities. There is a significant clinical need for alternative therapy options and combinations. At the same time, the prognosis is in general severely limited and many patients are in low general condition, necessitating tolerability of potential therapeutic modalities. Therefore, the compelling pre-clinical rationale of hyperthermia as a radio- and chemosensitizer and its demonstrated clinical benefit in breast cancer substantiate its role as a combination candidate for the treatment of recurrent tumors.

The phase III study by Emami et al. [35] suffered from suboptimal quality assurance and compliance with the HT treatments, which therefore precluded a definitive statement on the efficacy of HT in this indication. The necessity to address these issues has been well known in the community. Quality assurance (QA) guidelines from expert commissions exist for several hyperthermia modalities in order to ensure adequate HT implementation and delivery [85,86,87,88], as well as more robust data in clinical studies. Additionally, it has to be mentioned that this trial only investigated recurrent lesions for which brachytherapy was feasible. Larger recurrent lesions that need treatment with fractionated external beam radiotherapy might be much better suited for additional hyperthermia, as ablative radiotherapy doses are usually not applicable in this more advanced setting.

As expected, the evidence level of the presented studies was low, as only one phase III randomized control study was identified and more than half of the studies were single-institution retrospective analyses. However, the aim of this review was primarily to provide an overview of the currently available literature, as well as to open up promising directions for further studies.

In rectal cancer, the high pCR rate in LRRC patients in the study by Ott et al. [46] was notable. The currently recruiting CAO/ARO/AIO-16 study (ClinicalTrials.gov Identifier: NCT03561142) [89] investigates total neoadjuvant therapy in LARC patients with an option for additional regional hyperthermia. However, as patients with prior pelvic irradiation are excluded, further exploration of combined HT and re-RT in these patients should be performed to validate the previous result.

The dose de-escalation concept by Notter et al. [52] for RAS patients akin to patients with BC recurrence presents effective tumor control with a low radiation dose and the possibility for re-treatments with acceptable toxicity. For other entities that are prone to rapid recurrence and for which the benefit in tumor response conferred by additional hyperthermia can be established, a similar re-treatment concept may be considered.

Another indication to combine re-irradiation and hyperthermia for certain patients or tumor entities may be derived from genomic data, such as mRNA-based hypoxia signatures [90]. These may be utilized to explore re-RT and HT for tumor entities with a high average hypoxia score, such as HN cancer, or in an individualized, entity-agnostic fashion, based on biopsy data.

The currently recruiting HETERERO study (ClinicalTrials.gov Identifier: NCT04889742) [91] is a single-institution, multi-entity study for patients treated with HT and re-RT for locoregionally recurrent tumors. The aim is to demonstrate non-inferiority in time to local failure from the re-RT treatment compared to the initial RT.

Ultimately, further studies on re-RT and HT should build on the existing evidence available for breast cancer and plan further studies upon careful stratification and analysis of disease stages and treatment results. A consensus on patient selection and radiotherapy as well as hyperthermia treatment concepts (e.g., dosage, technique) may allow for prospective multi-institutional registry studies as an alternative to randomized controlled trials.

## 5. Conclusions

Outside of breast cancer, the evidence level for combined re-RT and HT is scarce, despite the compelling pre-clinical rationale and unmet clinical need. Uniform recurrence treatment recommendations do not exist for most cancers, despite the increasing life expectancy of cancer survivors. Further clinical trials are required to demonstrate the benefit of added hyperthermia.

## Figures and Tables

**Figure 1 cancers-15-00742-f001:**
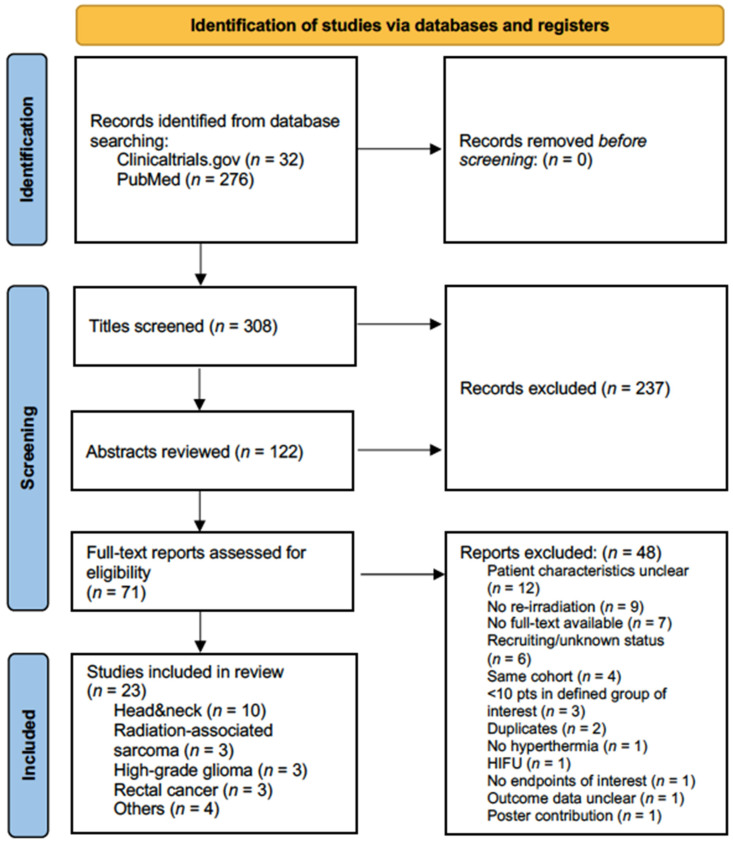
PRISMA flow diagram of the literature search.

**Table 1 cancers-15-00742-t001:** Overview of included studies for combined re-irradiation and hyperthermia treatment.

Author	Year	Entity Group	Type of Study	*N*	Treatment	Type of HT	COI	Funding
Petrovich et al. [34]	1989	HN	Single institution, prospective phase I/II, single arm	20	Interstitial BT + iHT	Interstitial, MW	n.r.	n.r.
Emami et al. [35]	1996	HN	Multi-center, prospective, randomized phase III	40	Interstitial BT + iHT vs. interstitial BT	Interstitial, MW/RF	n.r.	n.r.
Feyerabend et al. [36]	1997	HN	Single institution, prospective phase I/II, single arm	13	EBRT + HT + CT	Superficial, radiative, MW	n.r.	Grants by Deutsche Krebshilfe, Deutsche Forschungsgemeinschaft (DFG), Cancer Research Institute (NY, USA)
Puthawala et al. [37]	2001	HN	Single institution, prospective phase I/II, single arm	133	Salvage interstitial-LDR-BT + HT +/− CT	Interstitial MW	n.r.	n.r.
Geiger et al. [38]	2002	HN	Single institution, prospective phase I/II, single arm	15	Interstitial-PDR-BT + HT + CT	Interstitial, MW	n.r.	n.r.
Gabriele et al. [39]	2009	HN	Two-center, retrospective	14	EBRT + HT	Superficial, radiative, MW	none	none
Bartochowska et al. [40]	2012	HN	Two-center, retrospective	16	Palliative interstitial-HDR/PDR-BT + iHT	Interstitial, MW	n.r.	n.r.
Verduijn et al. [41]	2018	HN	Single institution, retrospective	18	RT(IMRT/CBK/BT) + HT +/− OP	Deep local, radiative, RF	Co-founders of Sensius BV	Sensius BV and KWF Kankerbestrijding (Dutch Cancer Society) grant
Kroesen et al. [42]	2021	HN	Single institution, retrospective	22	RT(CBK/IMRT/VMAT) + HT +/− OP	Deep local, radiative, MW	Co-founders of Sensius BV	Sensius BV and KWF Kankerbestrijding (Dutch Cancer Society) grant
Zschaeck et al. [43]	2021	HN	Single institution, prospective phase I	10	EBRT + FRWBH +/− CT +/− OP	Whole body, wIRA	none	Dr. med. h.c. Erwin Braun Stiftung
Juffermans et al. [44]	2003	REC	Single institution, retrospective	54	EBRT + HT	Deep regional, radiative, RF	n.r.	n.r.
Milani et al. [45]	2008	REC	Single institution, prospective phase I/II	24	EBRT + HT + CT	Deep regional, radiative, RF	n.r.	n.r.
Ott et al. [46]	2021	REC	Multi-center, prospective, phase I/II	10	EBRT + HT + CT	Deep regional, radiative, RF	none	none
Maier-Hauff et al. [47]	2007	CNS	Single institution, prospective, single arm	11	EBRT + HT	Internal, Fe3O4 (magnetite nanoparticles)	Employee/co-founder of MagForce AG, patents	n.r.
Maier-Hauff et al. [48]	2011	CNS	Prospective, single arm, two-center phase II	59	EBRT + HT	Internal, Fe3O4 (magnetite nanoparticles)	Employee/co-founder of MagForce AG, patents	MagForce AG
Heo et al. [49]	2017	CNS	Single institution, retrospective	20	EBRT + HT +/− CT, OP	External, capacitive, RF	n.r.	n.r.
de Jong et al. [50]	2012	RAS	Two-center, retrospective	16	EBRT + HT	Superficial, radiative, MW	none	none
Linthorst et al. [51]	2013	RAS	Two-center, retrospective	24	EBRT + HT +/− OP	Superficial, radiative, MW	none	n.r.
Notter et al. [52]	2021	RAS	Multi-center, retrospective	10	EBRT + HT	Superficial, wIRA	none	none
Surwit et al [53]	1983	CER	Single institution, prospective phase I	12	Interstitial-LDR-BT + iHT	Interstitial, RF	n.r.	n.r.
Gupta et al. [54]	1999	ENDO	Single institution, retrospective	15	Interstitial-LDR-BT + iHT	Interstitial, RF	n.r.	n.r.
Yamaguchi et al. [55]	2011	ESO	Single institution, retrospective	14	EBRT + HT	Deep, capacitive, RF	none	n.r.
Ohguri et al. [56]	2012	LU	Single institution, retrospective	33	EBRT + HT	Deep, capacitive, RF	none	n.r.

Abbreiations: BT = brachytherapy, CBK = cyberknife, CER = cervical cancer, COI = conflict of interest, CNS = central nervous system, CT = chemotherapy, EBRT = external beam radiotherapy, ENDO = endometrial cancer, ESO = esophageal cancer, FRWBH = fever-range whole-body hyperthermia, HN = head and neck cancer, HT = hyperthermia, iHT = interstitial hyperthermia, IMRT = intensity-modulated radiotherapy, LDR = low dose rate, LU = lung cancer, MW = microwave, n.r. = not reported, OP = operation, PDR = pulsed dose rate, RAS = radiation-associated sarcoma, REC = rectal cancer, RF = radiofrequency, RT = radiotherapy, VMAT = volumetric modulated arc therapy, wIRA = water-filtered infrared-A, wk = week.

**Table 2 cancers-15-00742-t002:** Head and neck cancer.

Author	Year	Study Type	*N*	Entity	Treatment	Re-RT Dose (Mean/Median(Range), Gy)	HT	Response	Toxicity
Petrovich et al. [34]	1989	Single institution, prospective phase I/II, single arm	20	HN	Interstitial BT + iHT	40 Gy or 50 Gy dep. on prior RT	Interstitial, MW	CR 68%, PR 32%, median OS 8.5 mo, 2y-OS 18%, 95% palliation	Acute: G3 aspiration pneumonia (*n* = 1), G4 soft tissue necrosis (*n* = 1)
Emami et al. [35]	1996	Multi-center, prospective, randomized, phase III, two arms, multiple entities, 75/176 HN	40 ITRT vs. 35 IRT (176), 84% re-RT	HN	Interstitial BT + iHT vs. interstitial BT	n.r.	Interstitial, MW/RF	CR 62% vs. 52% n.s., PR 4% vs. 13% n.s., 2y-LC (43%) vs. (37%) n.s.	Acute: ≥G3 22% vs. 12%, G4 10% vs. 3% (skin/subcutaneous/mucosal), late: ≥G3 20% vs. 15% n.s.
Feyerabend et al. [36]	1997	Single institution, prospective phase I/II, single arm	13	HN	EBRT + HT + CT	36(30–50)	Superficial, radiative, MW	CR 8%, PR 84%	Acute: G3 skin reaction (*n* = 1)
Puthawala et al. [37]	2001	Single institution, prospective phase I/II, single arm, 133/220 with HT	133 (220)	HN	Salvage interstitial-LDR-BT + iHT +/− CT	53(35–65)	Interstitial, MW	CR (77%), 2y-LC (69%), +/− HT n.s.	n.r.
Geiger et al. [38]	2002	Single institution, prospective phase I/II, single arm	15	HN	Interstitial-PDR-BT + iHT + CT	55(34–60)	Interstitial, MW	2y-LC 68%, 2y-OS 67%	Acute: G3 soft tissue ulceration (*n* = 1)
Gabriele et al. [39]	2009	Two-center, retrospective, multiple entities, 14/51 HN	14 (51)	HN	EBRT + HT	n.r.	Superficial, radiative, MW	CR 33%, PR 25%, NR 41.7%, 18-mo-LC 50%	No acute/late ≥G3 toxicity observed
Bartochowska et al. [40]	2012	Two-center, retrospective, 16/156 with HT	16 (156)	HN	Palliative interstitial-HDR/PDR-BT + iHT	HDR (12–20), PDR 20(20–40)	Interstitial, MW	Median OS: (7 mo), 2y-OS: (17%), +/− HT n.s.	No excess toxicity +/− HT
Verduijn et al. [41]	2018	Single institution, retrospective, 18/27 with re-RT	18 (27)	HN	RT(IMRT/CBK/BT) + HT +/− OP	IMRT (40–70 Gy/2 Gy), CBK (5 × 5.5 Gy, 6 × 5 Gy, 6 × 5.5 Gy or 6 × 6 Gy 2×/wk), BT (38 Gy in 12 Fx)	Deep local, radiative, RF	CR 39%, 2y-LC: 36%, 2y-OS: 33%	Tube feeding (*n* = 11), radiation dermatitis (*n* = 2), pneumonitis (*n* = 2), fibrosis (*n* = 1) for all patients
Kroesen et al. [42]	2021	Single institution, retrospective	22	HN	RT(CBK/IMRT/VMAT) + HT +/− OP	IMRT/VMAT 60(20–60), CBK (6 × 5.5 Gy)	Deep local, radiative, MW	2y-LC: 36.4%, 2y-OS 54.6% after definitive therapy (11/22)	lLte: ≥G3 39.2% (xerostomia, dysphagia, osteoradionecrosis and trismus)
Zschaeck et al. [43]	2021	Single institution, prospective phase I	10	HN	EBRT + FRWBH +/− CT	66 Gy/1.2 Gy bi-daily	FRWBH wIRA	Completion rate 50%, median OS 10 mo	No increased toxicity with more HT sessions

Abbreviations: BT = brachytherapy, CBK = cyberknife, CR = complete response, CT = chemotherapy, EBRT = external beam radiotherapy, FRWBH = fever-range whole-body hyperthermia, HN = head and neck cancer, HT = hyperthermia, iHT = interstitial hyperthermia, IMRT = intensity-modulated radiotherapy, ITRT = interstital thermoradiotherapy, IRT = interstitial radiotherapy, LC = local control, LDR = low dose rate, mo = months, MW = microwave, n.r. = not reported, n.s. = not significant, OP = operation, OS = overall survival, PDR = pulsed dose rate, PR = partial response, RF = radiofrequency, RT = radiotherapy, VMAT = volumetric modulated arc therapy, wIRA = water-filtered infrared-A, wk = week.

**Table 3 cancers-15-00742-t003:** Glioma.

Author	Year	Study Type	*N*	Entity	Treatment	Re-RT Dose (Mean/Median(Range), Gy)	HT	Response	Toxicity
Maier-Hauff et al. [47]	2007	Single institution, prospective, single arm, 11/14 with re-RT	11 (14)	CNS (GBM)	EBRT + HT +/− adjuvant CT	(20–30)	Internal, Fe3O4 (magnetite nanoparticles)	Median OS: 14.5 mo from primary diagnosis, 7.6 mo “after reintervention”	No treatment-related toxicity observed
Maier-Hauff et al. [48]	2011	Prospective, single arm, two-center phase II	59	CNS (GBM)	EBRT + HT	30 Gy/2 Gy	Internal, Fe3O4 (magnetite nanoparticles)	Median OS: 13.4 mo from recurrence diagnosis	23.7% seizures, 21% motor disturbances
Heo et al. [49]	2017	Single institution, retrospective	20	CNS (HGG, 80% III, 20% IV)	EBRT + HT +/− CT, OP	30(16–40)	External, capacitive, RF	Median OS: 8.4 mo from start of re-RT	No ≥G3 toxicity reported during treatment

Abbreviations: CNS = central nervous system, CT = chemotherapy, EBRT = external beam radiotherapy, GBM = glioblastoma, HGG = high-grade glioma, HT = hyperthermia, mo = months, OP = operation, OS = overall survival, RF = radiofrequency.

**Table 4 cancers-15-00742-t004:** Radiation-associated sarcoma.

Author	Year	Study Type	N	Entity	Treatment	Re-RT Dose (Mean/Median(Range), Gy)	HT	Response	Toxicity
de Jong et al. [50]	2012	Two-center, retrospective	16 (13 unresectable, 3 surgery)	RAS	EBRT + HT +/− OP	32(6–36)	Superficial, radiative, MW	Median OS: 9 mo; unresectable: 3y-LC: 31%	Late: G4 peripheral limb ischemia (*n* = 1)
Linthorst et al. [51]	2013	Two-center, retrospective	24 (13 unresectable, 11 surgery)	RAS	EBRT + HT +/− OP	32(32–54)	Superficial, radiative, MW	Median OS: 12 mo; unresectable: OS 5 mo, 3y-LC: 22%; surgery: OS 13 mo, 3y-LC: 46%	Acute: G3 wound infection (*n* = 1), late: G4 osteonecrosis (*n* = 1), G4 chronic wound (*n* = 1)
Notter et al. [52]	2021	Multi-center, retrospective	10	RAS	EBRT + wIRA HT	5 × 4 Gy 1 Fx/wk, 1 patient 25 × 2 Gy 5 Fx/wk	Superficial, wIRA	Median OS: 17 mo	No acute/late ≥G3 toxicity observed

Abbreviations: EBRT = external beam radiotherapy, HT = hyperthermia, LC = local control, mo = months, MW = microwave, OP = operation, OS = overall survival, RAS = radiation-associated sarcoma, wIRA = water-filtered infrared-A, wk = week.

**Table 5 cancers-15-00742-t005:** Rectal cancer.

Author	Year	Study Type	*N*	Entity	Treatment	Re-RT Dose (Mean/Median(Range), Gy)	HT	Response	Toxicity
Juffermans et al. [44]	2003	Single institution, retrospective	54	REC	EBRT + HT	32(24–32)	Deep regional, radiative, RF	Completion rate 87%, median OS 10 mo, palliative effect 83%, duration of palliation 6 mo	No acute/late ≥G3 toxicity observed
Milani et al. [45]	2008	Single institution, prospective phase I/II	24	REC	EBRT + HT + CT	39.6(30–45)	Deep regional, radiative, RF	Completion rate 92%, palliative effect 70% of responding patients, 3y-LPFS 15%, median OS 27 mo, 3y-OS 30%	Acute G3 diarrhea 12.5% of all patients
Ott et al. [46]	2021	Multi-center, prospective, phase II, LARC and LRRC, 10/16 LRRC with re-RT	10 (16)	REC	EBRT + HT + CT +/− OP	45 Gy/1.8 Gy	Deep regional, radiative, RF	Completion rate RT (99%), HT (90%), LRRC: 3y-LPFS (49%), 3y-OS (85%), pCR (19%)	G3 toxicity n.r. for LRRC patients separately, no G4/5 toxicity observed

Abbreviations: CR = complete response, CT = chemotherapy, EBRT = external beam radiotherapy, HT = hyperthermia, LARC = locally advanced rectal cancer, LC = local control, LPFS = local progression-free survival, LRRC = locally recurrent rectal cancer, mo = months, n.r. = not reported, n.s. = not significant, OP = operation, OS = overall survival, pCR = pathologic complete response, PR = partial response, REC = rectal cancer, RF = radiofrequency, RT = radiotherapy.

**Table 6 cancers-15-00742-t006:** Other cancer types.

Author	Year	Study Type	*N*	Entity	Treatment	Re-RT Dose (Mean/Median(Range), Gy)	HT	Response	Toxicity
Surwit et al. [53]	1983	Single institution, prospective phase I	21	12 cervical, 2 vaginal/urethral, 3 uterine, 4 ovarian	Interstitial-LDR-BT + iHT	22(15–45.6)	Interstitial RF	CR+PR 81%, duration of response 4 mo, palliation effect 76% among patients with tumor response	Fistulae in 4/21 patients
Gupta et al. [54]	1999	Single institution, retrospective, 15/69 with re-RT	15 (69)	10 endometrial, 2 vaginal/urethral, 3 cervical	Interstitial-LDR-BT + iHT	35(25–55)	Interstitial RF	3y-LC 49%	G4: (14%), no excess toxicity +/− HT
Yamaguchi et al. [55]	2011	Single institution, retrospective, 14/31 with HT	14 (31)	Esophageal	EBRT (3D-CRT) + HT +/− CT	40 Gy (curative), 36 Gy (palliative)	Deep, capacitive, RF	median OS: 8.1 mo, +/− HT n.s.	≥G3 esophageal complications: 6/31 patients
Ohguri et al. [56]	2012	Single institution, retrospective	33	NSCLC	EBRT+HT	50(29–70)	Deep, capacitive, RF	RT completion rate 97%, median OS: 18.1 mo, DLC: 12.1 mo, PFS: 6.7 mo	≥G3: acute: thrombocytopenia (*n* = 1), pleuritis (*n* = 1); late: brachial plexus neuropathy (*n* = 1)

Abbreviations: 3D-CRT = 3D conformal radiation therapy, BT = brachytherapy, CR = complete response, CT = chemotherapy, DLC = duration of local control, EBRT = external beam radiotherapy, HT = hyperthermia, iHT = interstitial hyperthermia, LC = local control, LDR = low dose rate, mo = months, n.s. = not significant, NSCLC = non-small-cell lung cancer, OS = overall survival, PFS = progression-free survival, PR = partial response, RF = radiofrequency, RT = radiotherapy.

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
