# Peer review of "Combined Hyperthermia and Re-Irradiation in Non-Breast Cancer Patients: A Systematic Review"

_cancers, 2023, doi:10.3390/cancers15030742_

Round 1

Reviewer 1 Report

In the text, a lot of different hyperthermia types are mentioned; it will be good to describe them - what is common between them, what is different, possible mechanism of action, source, etc

The "meta - analysis" is basically a summary of clinical trials on the topic, and it will be helpful to provide some general statistical analysis. 

I do not understand why "non-breast cancer" is mentioned in the title. 

Author Response

We would like to thank the reviewer for their thoughtful comments and efforts towards improving our manuscript. In the following, we will address each comment and suggestion specifically by this reviewer.

We agree that the review includes various applicators of hyperthermia with different functional principles. We included an additional paragraph which I. outlines the different modalities and II. Describes their similarities and differences. We hope this issue to be adequately addressed.

The reviewer moreover points out that the meta-analysis in this review has been a summary of the clinical trials at hand and suggests general statistical analyses to be added. We agree that summaries of current clinical studies would be more insightful in general with appropriate statistical analyses to provide quantitative values on certain endpoints. However, this necessitates the summarized clinical trials to be sufficiently homogeneous and provide data on the endpoints to enable such an analysis. In our case, there was no entity in which all included studies reported were comparable and reported the same endpoints data including a variability parameter. We therefore abstained from a meta-analysis, which we stated in the review. It is nonetheless interesting to evaluate the reasons for the heterogeneity of the studies, which we summarized in each respective paragraph in the “Results” section. We thank the reviewer for this justified argument, and we hope that the reasons stated above sufficiently explain the way we composed this manuscript.

Lastly, the reviewer expresses concern about the title of this review, more specifically why “non-breast cancer” has been added to describe the patients in question. As stated in the introductory paragraphs, there already have been previous systematic reviews in literature for breast-cancer patients undergoing combined re-irradiation and hyperthermia. We therefore did not include this patient group in our analysis. In accordance to the PRISMA guidelines we also included this information in the title to give a succinct overview of the clinical question at hand.

Reviewer 2 Report

Definition of specific aim is lacking. What is the merit of this paper?

Author Response

We thank the reviewer for the feedback. In the following we would like the issue addressed by the reviewer.

The primary concern voiced by this reviewer is that the aim of this systematic review has not been clear. We would express our regrets that the purpose and merit of this systematic review could not be elucidated to the reviewer after reading our manuscript. In short, it is our understanding that hyperthermia is an effective anti-cancer treatment [REFS 1-5], and that the pre-clinical rationale of radiosensitization as well as the unmet clinical need to alternative therapies for patients undergoing re-irradiation represent compelling arguments for this combination treatment. Except in the case of breast cancer, there has so far not been a literature review of studies summarizing the available evidence on re-irradiation and hyperthermia. Therefore, the two aims of this review were to firstly summarize the available evidence as mentioned before, and secondly, to pinpoint through the analysis what kind of future studies may be sensible.

We have revised the respective paragraphs in the sections: Abstract, introduction, conclusion. We hope that the revisions have made the points given above clearer.

Reviewer 3 Report

The authors have executed a thorough evaluation on the research performed using hyperthermia in combination with  re-irradiation in non-breast cancer patients. The Review of the literature is performed according to the updated PRISMA guidelines and includes review of potential COI. The Review is comprehensive and well-described and -discussed by the authors. Most included studies suffer from lack of control groups, variation in patient selection, radiotherapy dosage and hyperthermia technique / QA. Therefore no sound conclusion can be obtained from this review, as the authors rightly conclude as well. 

Minor comments:

-in the conclusions, could the authors elaborate on how to improve future studies, bearing in mind that a randomized phase III trial is highly likely never to happen in the re-irradiation setting? My suggestions would be the running of prospective multi-institutional studies with a consensus on patient selection, radiotherapy tehcnique/scheme/dosage and hyperthermia technique/ dosage and QA. These kind of registries would also include patients that refuse hyperthermia as a control (of course resulting in bias, but as a best possible control group).

Author Response

We would like to thank the reviewer for their constructive comments. We have amended a paragraph in the “Discussions” about possible designs of future studies in combined hyperthermia and (re-)irradiation. We hope this issue to be thereby adequately addressed.

Round 2

Reviewer 1 Report

Below the hyperthermia delivery characterization (line 78) chapter, an additional paragraph will be needed to describe the biological mechanism of hyperthermia in the cancer tissue (for temperature range 39-43). This can help to understand why it can be beneficial for combinatory treatment.

The unfinished sentence at line 438.

Author Response

We would like to thank the reviewer for the helpful comment. The reviewer suggests that another paragraph which would outline the biological effects of mild hyperthermia would help contextualize the previous paragraph about the hyperthermia modalities, as well as help the reader understand the purpose of the combination treatment. We have added another paragraph, briefly outlining the biological effects of mild hyperthermia in the given temperature range. 

The unfinished sentence has been removed.